

# Comparison between Mother, ActiGraph wGT3X-BT, and a hand tally for measuring steps at various walking speeds under controlled conditions

Henrik Riel[1,2], Camilla Rams Rathleff[2], Pernille Møller Kalstrup[2], Niels Kragh Madsen[2], Elena Selmar Pedersen[2], Louise Bilenberg Pape-Haugaard[2] and Morten Villumsen[2,3]

[1] Research Unit for General Practice, Department of Clinical Medicine, Aalborg University, Aalborg, Denmark
[2] Department of Health Science and Technology, Aalborg University, Aalborg, Denmark
[3] Department of Physiotherapy, University College of Northern Denmark, Aalborg, Denmark

## ABSTRACT

**Introduction**. Walking is endorsed as health enhancing and is the most common type of physical activity among older adults. Accelerometers are superior to self-reports when measuring steps, however, if they are to be used by clinicians the validity is of great importance. The aim of this study was to investigate the criterion validity of Mother and ActiGraph wGT3X-BT in measuring steps by comparing the devices to a hand tally under controlled conditions in healthy participants.

**Methods**. Thirty healthy participants were fitted with a belt containing the sensor of Mother (Motion Cookie) and ActiGraph. Participants walked on a treadmill for two minutes at each of the following speeds; 3.2, 4.8, and 6.4 km/h. The treadmill walking was video recorded and actual steps were subsequently determined by using a hand tally. Wilcoxon's signed ranks test was used to determine whether Mother and ActiGraph measured an identical number of steps compared to the hand tally. Intraclass correlation coefficients were calculated to determine the relationship and Root Mean Square error was calculated to investigate the average error between the devices and the hand tally. Percent differences (PD) were calculated for between-instrument agreement (Mother vs. the hand tally and ActiGraph vs. the hand tally) and PDs below 3% were interpreted as acceptable and clinically irrelevant.

**Results**. Mother and ActiGraph under-counted steps significantly compared to the hand tally at all walking speeds ($p < 0.001$). Mother had a median of total differences of 9.5 steps (IQR = 10) and ActiGraph 59 steps (IQR = 77). Mother had smaller PDs at all speeds especially at 3.2 km/h (2.5% compared to 26.7%). Mother showed excellent ICC values $\geq$0.88 (0.51–0.96) at all speeds whilst ActiGraph had poor and fair to good ICC values ranging from 0.03 (−0.09–0.21) at a speed of 3.2 km/h to 0.64 (0.16–0.84) at a speed of 6.4 km/h.

**Conclusion**. Mother provides valid measures of steps at walking speeds of 3.2, 4.8, and 6.4 km/h with clinically irrelevant deviations compared to a hand tally while ActiGraph only provides valid measurements at 6.4 km/h based on the 3% criterion. These results have significant potential for valid objective measurements of low walking speeds. However, further research should investigate the validity of Mother in patients at even slower walking speeds and in free-living conditions.

Corresponding author
Henrik Riel, hriel@dcm.aau.dk

## INTRODUCTION

Physical activity (PA) is endorsed as health enhancing (*Steeves et al., 2015*) and is known to prevent and reduce both musculoskeletal disorders and mortality (*Holtermann et al., 2012*; *Holtermann et al., 2013*; *Haskell et al., 2007*; *Blair & Morris, 2009*). Additionally, in older adults, PA is especially important in maintaining self-dependence, preventing disease and improving the quality of life (*NHS Choices, 2015*). In contrast, the lack of PA is related to muscular alterations such as atrophy and decreased muscle strength (*Convertino, Bloomfield & Greenleaf, 1997*; *Appell, 1990*), thus possibly contributing to loss of self-dependence, especially in older adults. During hospitalization geriatric patients have shown low levels of PA (*Villumsen et al., 2015*) and only 17.8% of patients regain their pre-hospitalization level of mobility function 12 months after admission (*Visser et al., 2000*). This emphasizes the need for PA awareness.

Walking is the most common type of leisure-time PA among adults and the prevalence of walking for PA increases with age up until 65–74 years (*Rafferty et al., 2002*). In order to measure PA, accelerometers are superior and recommended with respect to validity and applicability (*Müller, Winter & Rosenbaum, 2010*) compared to self-reports, as patients often over- or underestimate their actual level of PA (*Sallis & Saelens, 2000*; *Farni et al., 2014*; *Barriera et al., 2013*). Even though studies have found the validity and specificity to be high when measuring different types of PA (i.e., moderate and high pace walking), accelerometers are considered inadequate when measuring steps at low walking speeds (*Crouter et al., 2013*; *Turner et al., 2012*; *Steeves et al., 2011*; *Webber et al., 2014*; *Dijkstra et al., 2008*; *Barriera et al., 2013*). If accelerometers are to be successfully used by patients and/or healthcare personnel it may be of importance that the accelerometers are valid, versatile, user-friendly, and inexpensive.

One of the most commonly used accelerometers for monitoring PA is ActiGraph (ActiGraph, Pensacola, FL, USA) (*Crouter et al., 2013*; *Barriera et al., 2013*; *Ekblom et al., 2012*; *Herman Hansen et al., 2014*). However, ActiGraph is developed with the intention to be used by physicians and in research (Pensacola, FL, USA) whilst a new accelerometer, Mother (Sen.se, Paris, France), is developed with the intention to be used by the private consumer (Sen.se, Paris, France). The aim of this study was to investigate the criterion validity of Mother (Sen.se, Paris, France) and ActiGraph wGT3X-BT (Pensacola, FL, USA) in measuring steps by comparing the devices to a hand tally, which is considered gold standard, under controlled conditions in healthy participants.

## METHODS

This study is a validity study that complies with the Guidelines for Reporting Reliability and Agreement Studies (GRRAS) (*Kottner et al., 2011*).

| Table 1 Participant demographics. | |
|---|---|
| Gender (N, men/women) | 15/15 |
| Age (years, mean (SD)) | 27.9 (±4.2) |
| Height (cm, mean (SD)) | 173.5 (±9.1) |
| Weight (kg, mean (SD)) | 71.6 (±11.3) |
| BMI (kg/m$^2$, mean (SD)) | 23.6 (±2.2) |

## Ethics statement

Ethical approval of the research protocol was not needed according to The North Denmark Region Committee on Health Research Ethics. Written informed consent was signed by all participants prior to the study.

## Sample size and raters

Sample size was determined to be 30 participants using large sample case (*Hogg & Tanis, 1996*). Two raters performed the hand tallying and conducted the treadmill test whilst two different raters performed the data treatment without being part of the data collection.

## Participants

Thirty-one healthy students were recruited from Aalborg University, Denmark (male $n = 15$, female $n = 16$). Data was collected from March to April 2015.

The inclusion criteria were: (i) age of 18 years or above, (ii) no self-reported health problems evaluated by the Physical Activity Readiness Questionnaire (PAR-Q) (*Shephard, 1988*), (iii) ability to walk without walking aids, (iv) ability to walk continuously for 10 min on a treadmill, and (v) ability to read, understand and speak Danish and English. The fifth criterion was chosen to ensure that participants understood the instructions and the PAR-Q, which was in English, as no Danish translation was available.

The exclusion criteria were: (i) pregnancy (self-reported), (ii) BMI $\geq$ 30 kg/m$^2$, and (iii) neurological diseases (self-reported).

Thirty-one healthy participants were recruited for the study. One participant was excluded due to a BMI > 30 kg/m$^2$. Participants did not report any impairments or morbidities potentially interfering with the assessment. See Table 1 for participant demographics.

## Mother

Mother (Sen.se, Paris, France) is a triaxial accelerometer released in March 2014 by Sen.se (Paris, France). The device consists of a hub (Mother), up to 24 sensors (Motion Cookies), and a software application (the Senseboard) (Sen.se, Paris, France). User access is gained through the Senseboard, which is a collection of different applications developed by Sen.se. The dimensions of a Motion Cookie are 5.0*2.2*0.4 cm with a weight of 6 grams (Sen.se, Paris, France). The sample rate is fixed at 25 Hz. The accelerometer has a dynamic range of ±2G and a precision of 12 bit (*Alain Romanet, e-mail correspondence with Sen.se, March 9th 2016*). In this study the application Walk (Sen.se, Paris, France) was selected for measuring steps. Data from the Motion Cookie is uploaded to Mother every 5 min (*Franck Biehler, e-mail correspondence with Sen.se, March 23rd 2015*) in fractions of varying durations.

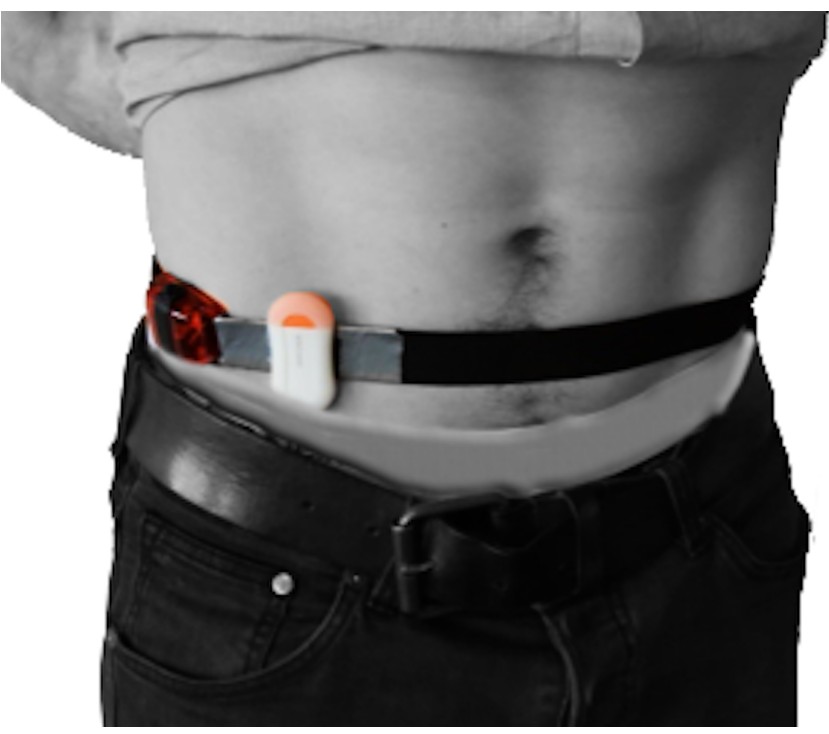

**Figure 1** **Accelerometer placement.** ActiGraph is placed laterally to the right SIAS and the Motion Cookie is placed medially (Randomization 1).

## ActiGraph wGT3X-BT

ActiGraph wGT3X-BT (ActiGraph LLC, Pensacola, FL, USA) is a triaxial accelerometer and one of the most commonly used devices for assessing PA (*Crouter et al., 2013*). The dimensions of the sensor are 4.6*3.3*1.5 cm with a weight of 19 grams. A sample rate of 100 Hz was chosen (range 30–100 Hz). The accelerometer has a dynamic range of $\pm 8G$ and a precision of 12 bit (ActiGraph LLC, Pensacola, FL, USA). Data is accessible by using the ActiLife Pro 6 software (ActiGraph LLC, Pensacola, FL, USA). After the recordings the sensor was connected to a computer through a mini-USB cable in order to upload the data. During initialization, information including subject name, gender, height, weight, race, limb and leg dominance is required, however random values were used as step counting is not affected by this information (ActiGraph LLC, Pensacola, FL, USA).

## Accelerometer placement

The participants wore Mother's Motion Cookie and ActiGraph simultaneously placed on an elastic belt above the right anterior superior iliac spine (SIAS) (0.5 cm. medially and laterally from the right SIAS, respectively) (Fig. 1). Hip placement has previously been found as the most precise single location placement of an accelerometer (*Cleland et al., 2013*). The placement of devices was randomized using a random number generator (http://www. random.org) (Randomization 0: Mother laterally, Randomization 1: ActiGraph laterally) to take possible placement related differences in validity into account. Randomization 0 was received by 17 participants and Randomization 1 was received by 13 participants.

### Hand tally

A hand tally is considered the gold standard when measuring steps (*Dijkstra et al., 2008*; *Fortune et al., 2014*; *Stemland et al., 2015*). In this study, the application AGR Tally counter (ver. 1.0, Angel Garcia Rubio) was used for hand tallying. Steps were measured by tapping the screen of an iPhone 4s (Apple Inc., Infinite Loop Cupertino, California, USA).

### Procedures of the treadmill test

The number of steps was obtained during a treadmill test where participants walked on a treadmill for two minutes at each of the three walking speeds; 3.2, 4.8, and 6.4 km/h. The inclination was set to 0°. The treadmill was preprogrammed to standardize the procedure for an in- and decrease of the walking speed. The two minutes included the time the treadmill in- and decreased the walking speed. The speeds were chosen in accordance with walking speeds chosen in three previous studies that investigated the validity of measuring steps under controlled conditions at various walking speeds (*Steeves et al., 2011*; *De Cocker et al., 2012*; *Clemes et al., 2010*). The test was conducted in the sports science laboratory at Aalborg University, Denmark. To take the inability of Mother to synchronize more often than every 5 min into consideration, the participants were asked to stand still for five minutes and 10 s before and after each walking speed in order to identify the walking session in the application programming interface (API).

### Data Treatment

Data from Mother for each test were identified by examining the walking duration in the API. Even though the participants walked for exactly two minutes, which was confirmed by video recordings of the tests, walking sessions had durations ranging from 115–130 s. The output from the API showing the number of steps of the walking session was manually examined to identify equipment malfunctions such as missing steps. Accelerometer data from ActiGraph were downloaded using ActiLife 6 Pro software.

Video recordings of the treadmill test were used for hand tallying. The definition of a step was adopted by *Dijkstra et al. (2008)* and defined as "*the first moment at which the heel of the foot for the initial step cleared the ground and the moment at which the foot of the closing step made completely contact with the floor*". The hand tallying procedure was double validated as the two raters hand tallied independently. The two raters had 100% agreement. The two raters responsible for data treatment and the statistical analyses had not been involved in the treadmill test nor the hand tallying.

### Statistical analysis

All statistical analyses were performed using IBM SPSS (ver. 22, IBM Corporation, New York, United States) with a significance level of $p < 0.05$.

Normal distribution was examined based on the differences between the number of steps measured by Mother and the hand tally and ActiGraph and the hand tally using Q-Q plots and Shapiro–Wilk test. Q-Q plots were assessed and as data did not appear to be normally distributed, a Shapiro–Wilk test was performed and confirmed that data were non-normally distributed ($p < 0.05$) (DOI: 10.6084/m9.figshare.3814272.v1).

Wilcoxon's signed ranks test was used to create a pairwise comparison to determine whether the devices and the hand tally measured the same number of steps. Means and standard deviations (SD) were computed for age, height, weight and BMI whilst medians and interquartile ranges (IQR) were computed for steps and differences in steps measured by the devices. A two-way random effects model (2.1), single measures, absolute agreement, and intraclass correlation coefficients (ICC) with 95% confidence intervals were used to express interrater reliability between the devices and the hand tally. ICC values >0.75 were interpreted excellent, 0.40–0.75 were interpreted fair to good and <0.40 were interpreted poor (*Fleiss, 1999*). To investigate the average error Root Mean Square error (RMSe) was calculated between each device and the hand tally. As data were non-normally distributed the nonparametric approach to presenting Bland–Altman plots was adopted and the median, the 2.5th, and the 97.5th percentiles were visualised in the plots (*Gialamas et al., 2010*).

Percent differences (PD) were calculated for between-instrument agreement (Mother vs. the hand tally and ActiGraph vs. the hand tally). The PD was calculated as $\frac{Absolute\ difference}{hand\ tally\ steps} \cdot 100$. Any negative values were converted to a positive to calculate the absolute difference. Clinical relevance of potential under- or over-counting of steps by the two accelerometers compared to hand tally was determined using a 3% criterion, which was based on previous studies (*Johnson et al., 2015*; *Holbrook, Barreira & Kang, 2009*; *Colley et al., 2013*; *Liu et al., 2015*). PD ≤3% were considered clinically irrelevant.

## RESULTS

### Mother vs. the hand tally

The median of differences in steps between Mother and the hand tally at the different walking speeds were 2.5 steps (IQR = 5) at 3.2 km/h, 2 steps (IQR = 2) at 4.8 km/h and 3.5 steps (IQR = 6) at 6.4 km/h (Table 2). These are depicted in the Bland–Altman plot (Fig. 2). The median of total differences was 9.5 steps (IQR = 10).

Wilcoxon's signed ranks test showed a significant difference in the number of steps measured by the hand tally vs. Mother at all walking speeds ($p < 0.001$) (Table 3).

The ICCs for Mother and the hand tally were all excellent ranging from 0.88 (0.51–0.96) at a speed of 3.2 km/h to 0.96 (0.72–0.99) at a speed of 4.8 km/h (Table 3). The RMSe ranged from 2.86 at 4.8 km/h to 5.50 at 3.2 km/h (Table 3). Mother had PDs ≤2.5% of the steps measured by the hand tally at all speeds (Table 3).

### ActiGraph vs. the hand tally

The median of differences in steps between ActiGraph and the hand tally at the different walking speeds were 49.5 steps (IQR = 69) at 3.2 km/h, 4 steps (IQR = 5) at 4.8 km/h and 4 steps (IQR = 5) at 6.4 km/h (Table 2). These are depicted in the Bland–Altman plot (Fig. 3). The median of total differences was 59 steps (IQR = 77).

Wilcoxon's signed ranks test showed a significant difference in the number of steps measured by the hand tally vs. ActiGraph at all walking speeds ($p < 0.001$) (Table 3).

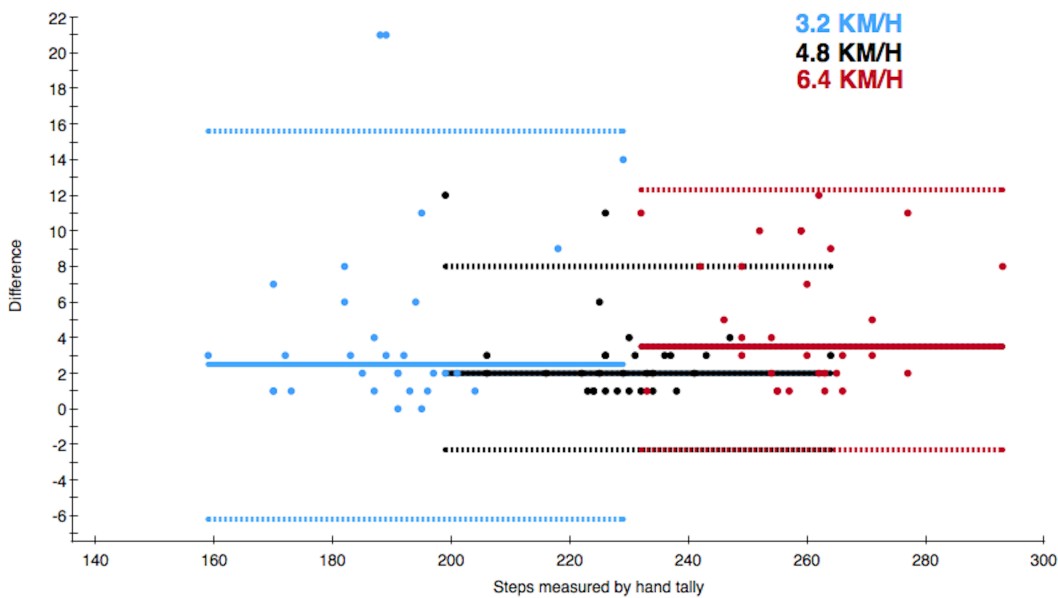

**Figure 2  Bland–Altman style plot of differences between the actual number of steps and steps measured by Mother.** The solid line depicts the median of differences and the dotted lines depict the 2.5th and 97.5th percentiles of each walking speed. The colour of the data point refers to the walking speed.

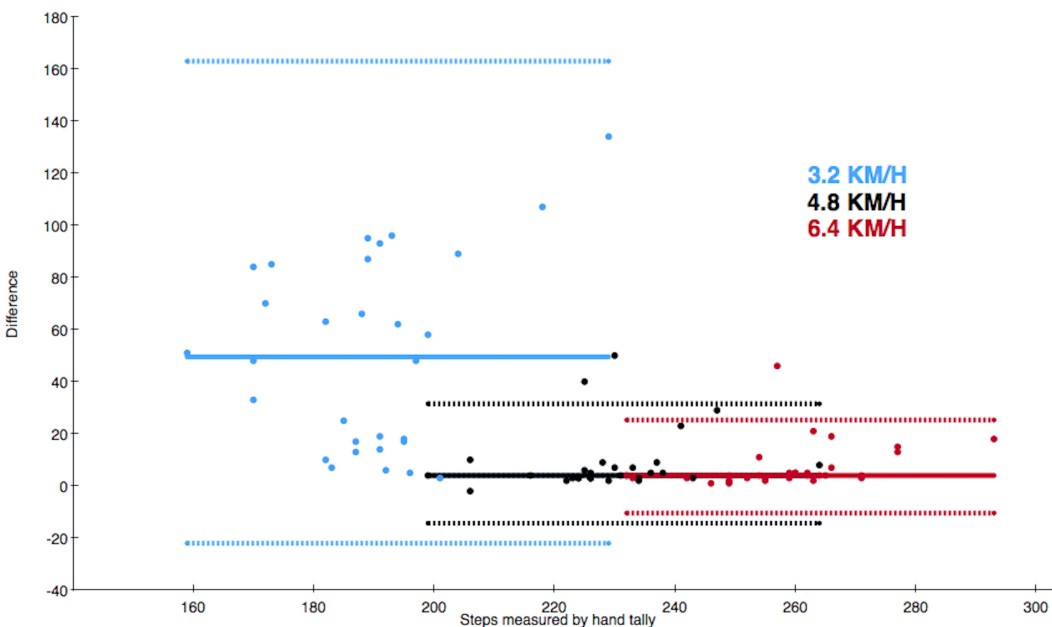

**Figure 3  Bland–Altman style plot of differences between the actual number of steps and steps measured by ActiGraph.** The solid line depicts the median of differences and the dotted lines depict the 2.5th and 97.5th percentiles of each walking speed. The colour of the data point refers to the walking speed.

**Table 2  The median of the number of steps measured by each device and the median of differences between the hand tally vs. Mother and the hand tally vs. ActiGraph.**

| | Walking speed (km/h) | Median of steps | Hand tally Median of differences in steps |
|---|---|---|---|
| Hand tally | 3.2 | 190 (IQR = 13) | – |
| | 4.8 | 229 (IQR = 11) | – |
| | 6.4 | 260 (IQR = 14) | – |
| | Total | 677 (IQR = 35) | – |
| Mother | 3.2 | 186 (IQR = 22) | 2.5 (IQR = 5) |
| | 4.8 | 227 (IQR = 12) | 2 (IQR = 2) |
| | 6.4 | 254 (IQR = 17) | 3.5 (IQR = 6) |
| | Total | 663 (IQR = 43) | 9.5 (IQR = 10) |
| ActiGraph | 3.2 | 134.5 (IQR = 70) | 49.5 (IQR = 69) |
| | 4.8 | 222 (IQR = 10) | 4 (IQR = 5) |
| | 6.4 | 254 (IQR = 15) | 4 (IQR = 5) |
| | Total | 615 (IQR = 73) | 59 (IQR = 77) |

**Table 3  Results from Wilcoxon's signed ranks test with significance levels of each comparison, intraclass correlation coefficient, Root Mean Square error, and percent difference.**

| | Walking speed (km/h) | Mother vs. Hand tally | ActiGraph vs. Hand tally |
|---|---|---|---|
| Wilcoxon's signed ranks test | 3.2 | $p < 0.001$[a] | $p < 0.001$[a] |
| | 4.8 | $p < 0.001$[a] | $p < 0.001$[a] |
| | 6.4 | $p < 0.001$[a] | $p < 0.001$[a] |
| | Total | $p < 0.001$[a] | $p < 0.001$[a] |
| Intraclass correlation coefficient (ICC (95 % CI)) | 3.2 | 0.88 (0.51–0.96) | 0.03 (−0.09–0.21) |
| | 4.8 | 0.96 (0.72–0.99) | 0.55 (0.13–0.78) |
| | 6.4 | 0.89 (0.19–0.97) | 0.64 (0.16–0.84) |
| | Total | 0.93 (0.18–0.98) | 0.22 (−0.10–0.54) |
| RMSe | 3.2 | 5.50 | 36.52 |
| | 4.8 | 2.86 | 11.66 |
| | 6.4 | 3.88 | 8.80 |
| | Total | 8.33 | 48.18 |
| PD (%) | 3.2 | 2.5 | 26.7 |
| | 4.8 | 1.3 | 3.7 |
| | 6.4 | 1.9 | 2.8 |
| | Total | 1.8 | 9.8 |

**Notes.**
[a]Significant difference.

The ICCs for ActiGraph and the hand tally were poor and fair to good ranging from 0.03 (−0.09–0.21) at a speed of 3.2 km/h to 0.64 (0.16–0.84) at a speed of 6.4 km/h (Table 3). The RMSe ranged from 8.80 at 6.4 km/h to 36.52 at 3.2 km/h (Table 3).

ActiGraph had PDs ≤26.7% of the steps measured by the hand tally (Table 3). ActiGraph was under-counting based on the 3% criterion at both 3.2 and 4.8 km/h.

## DISCUSSION

This study aimed at investigating the criterion validity of Mother and ActiGraph in measuring steps by comparing the devices to a hand tally under controlled conditions in healthy participants. The results revealed that both Mother and ActiGraph under-counted steps significantly compared to the hand tally at all walking speeds and Mother had smaller PDs at all walking speeds compared to ActiGraph.

Both Mother and ActiGraph under-counted steps significantly at all walking speeds compared to the hand tally, which is considered gold standard of measuring steps. Hence, it is apparent that none of these accelerometers have the same level of precision as the hand tally. However, significant results are not always clinically relevant as it would be unrealistic to use a hand tally under free-living conditions. Therefore, a 3% clinically irrelevant deviation from the steps measured by the hand tally was adopted inspired by previous studies of validity (*Johnson et al., 2015*; *Holbrook, Barreira & Kang, 2009*; *Colley et al., 2013*; *Liu et al., 2015*). The PDs of ≤2.5% measured by Mother were interpreted as being clinically irrelevant as it was less than 3%, however, ActiGraph had clinically relevant deviations at both 3.2 and 4.8 km/h. Especially at 3.2 km/h with a PD of 26.7% ActiGraph may not be adequately precise to measure steps in patients with a low walking speed. These results are in accordance with previous findings of the validity of ActiGraph at measuring steps at low walking speeds, which found that ActiGraph measured only 77.5% of the actual steps at the speed of 3.2 km/h (*Connolly et al., 2011*). In a practical perspective, ActiGraph would under-count 4,725 steps in a week if a patient has a daily average of 3,000 steps which would make it difficult for the clinician to determine whether or not the patient was following the advised PA. Accelerometers in general are known for being inadequate when measuring steps at the low speeds that some of the patients may walk at (*Barriera et al., 2013*; *Crouter et al., 2013*; *Turner et al., 2012*; *Steeves et al., 2011*; *Webber et al., 2014*; *Dijkstra et al., 2008*), but even though Mother is an accelerometer developed with the intention to be used by private consumers, it showed a superior accuracy compared to the accelerometer most commonly used for measuring PA in research (*Crouter et al., 2013*).

The excellent ICC between Mother and the hand tally would make a prediction of a margin of error at a given number of steps feasible. This means that with any given number of steps, the amount of miscounted steps can be estimated, thus making measurements with Mother more valid even at a large step counts.

### Limitations

The participants of this study were a group of younger, healthy subjects who performed steps that were similar to the definition of a step by *Dijkstra et al. (2008)*. However, patients may walk asymmetrically or without a swing phase which might provide different results. Healthy participants were chosen, as the purpose of this study was to test the validity under controlled conditions. Therefore, the results cannot be directly applied to any given patient group.

The walking sessions in the API of Mother had varying durations ranging from 115 to 130 s. The reason for this variation is unknown, however, it implies that the participants have either stopped walking prior to the end of the two minutes or they have continued moving even though the treadmill had stopped. This was, however, not indicated by the video recordings. Another explanation for the 115 s walking sessions could be that Mother stopped measuring as the participant slowed down before coming to a halt. As Mother measured durations both shorter and longer than the 120 s the walking session lasted, the inconsistencies in duration may have evened out.

### Future work

This study investigated step measuring at a walking speed of 3.2 km/h as the lowest speed, but some patients may walk at an even slower pace, thus investigating the validity at lower walking speeds is highly relevant to determine the minimum speed at which Mother still provides measures of steps that have clinically irrelevant deviations from the actual number of steps. Future studies should also include testing in a semi-controlled environment and in free-living conditions and should also include participants with a larger BMI than included in this study as the waist circumference can influence precision due to tilting (*Crouter, Schneider & Bassett, 2005*).

## CONCLUSION

Mother provides valid measures of steps at walking speeds of 3.2, 4.8, and 6.4 km/h with clinically irrelevant deviations compared to a hand tally while ActiGraph only provides valid measurements at 6.4 km/h based on the 3% criterion. These results have significant potential for valid objective measurements of low walking speeds.

### Funding
The authors received no funding for this work.

### Competing Interests
The authors declare there are no competing interests.

### Author Contributions
- Henrik Riel conceived and designed the experiments, performed the experiments, analyzed the data, contributed reagents/materials/analysis tools, wrote the paper, prepared figures and/or tables.
- Camilla Rams Rathleff and Pernille Møller Kalstrup conceived and designed the experiments, performed the experiments, contributed reagents/materials/analysis tools, reviewed drafts of the paper.
- Niels Kragh Madsen and Louise Bilenberg Pape-Haugaard conceived and designed the experiments, contributed reagents/materials/analysis tools, reviewed drafts of the paper.

- Elena Selmar Pedersen conceived and designed the experiments, performed the experiments, analyzed the data, contributed reagents/materials/analysis tools, reviewed drafts of the paper.
- Morten Villumsen wrote the paper, reviewed drafts of the paper.

## Human Ethics

The following information was supplied relating to ethical approvals (i.e., approving body and any reference numbers):

The North Denmark Region Committee on Health Research Ethics. It was stated by the committee that this study was not a health science research project according to their committee law and did not need approval.

## Data Availability

Riel, Henrik (2016): Data_final_Mother_ActiGraph.xlsx. figshare.
https://dx.doi.org/10.6084/m9.figshare.3814272.v1.

## Supplemental Information

Supplemental information for this article can be found online at http://dx.doi.org/10.7717/peerj.2799#supplemental-information.

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
