# Peer review of "Comparison between Mother, ActiGraph wGT3X-BT, and a hand tally for measuring steps at various walking speeds under controlled conditions"

_PeerJ, doi:10.7717/peerj.2799_

## Round 0.1 · original submission · Minor Revisions

This is an updated and revised version of your manuscript which was previously rejected after peer review. The previous Editor declined to handle this resubmission, so I was assigned to handle it.

The two reviewers (one of whom reviewed the original submission) and I believe that the revised manuscript is much improved; although one reviewer still has a few minor corrections requested. Please make these minor corrections so that the paper can be accepted for publication.

Reviewer 1 ·

Basic reporting

Overall, the authors have improved the structure and length of the manuscript. This has improved readability.

Experimental design

The authors have put much effort into re-analyzing their data. Also, I am pleased to see that 2 other walking speeds have been added. I am very content with the new analyses and reporting of the results.

Validity of the findings

The only comment I have, is that based on the introduction the reader may expect an older study population and a setting in which the tested walking speeds correspond better to the elderly population. Nonetheless, the authors do comment on this in the discussion and suggest that future research should focus on lower walking speeds

Additional comments

I am left with some minor comments only.

Line 106: ‘…of a Motion Cookie are 5*2.2*0.4’ It looks better to provide also a decimal following 5 (I assume it is 5.0).

Line 238-239: ‘The PDs of ≤2.5% measured by Mother was not interpreted as being clinically relevant as it was less than 3%, however’
I would suggest to write it as: …by Mother was interpreted as being clinically irrelevant… (otherwise it looks like you did not interpret the results)

Figure 2: the lower limit of the 6.4 km/h should be in red color.

·

Basic reporting

The manuscript is written well and easy to follow

Experimental design

The aim of the study is clear and the design is appropriate and well-explained.

Validity of the findings

The analyses are throrough and appropriate for the nature of the data. Selected speeds and thresholds are clearly justified. I am therefore confident the findings and subsequent conclusions are valid. The authors may also wish to state in the ‘Future Work’ that testing in overweight/obese subjects is warranted given that previous studies have noticed greater motion sensor inaccuracy in these groups (due to tilting).

Additional comments

Very well composed manuscript that will be of interest to researchers in the area.

---

## Round 0.2 · accepted · Accept

Thank you for making the requested revisions to this manuscript.

Reviewer 1 ·

Basic reporting

The authors did well to address all comments en suggestions as provided by the reviewers. I have no further comments.

Experimental design

Clear

Validity of the findings

Well presented